# Functional Aspects of Early Light-Induced Protein (ELIP) Genes from the Desiccation-Tolerant Moss *Syntrichia caninervis*

**DOI:** 10.3390/ijms21041411

**Published:** 2020-02-19

**Authors:** Xiujin Liu, Yigong Zhang, Honglan Yang, Yuqing Liang, Xiaoshuang Li, Melvin J. Oliver, Daoyuan Zhang

**Affiliations:** 1CAS Key Laboratory of Biogeography and Bioresource in Arid Land, Xinjiang Institute of Ecology and Geography, Urumqi 80031, China; liuxiujin14@mails.ucas.edu.cn (X.L.); zhangyigong203@163.com (Y.Z.); yanghonglan@ms.xjb.ac.cn (H.Y.); liangyuqing14@mails.ucas.edu.cn (Y.L.); lixs@ms.xjb.ac.cn (X.L.); 2Turpan Eremophytes Botanical Garden, Chinese Academy of Sciences, Turpan 838008, China; 3University of Chinese Academy of Sciences, Beijing 100049, China; 4USDA-ARS-Plant Genetic Research Unit, University of Missouri, Columbia, MO 65211, USA; Mel.Oliver@ARS.USDA.GOV

**Keywords:** ScELIP, high-light, photoprotection, photosynthesis

## Abstract

The early light-induced proteins (ELIPs) are postulated to act as transient pigment-binding proteins that protect the chloroplast from photodamage caused by excessive light energy. Desert mosses such as *Syntrichia caninervis*, that are desiccation-tolerant and homoiochlorophyllous, are often exposed to high-light conditions when both hydrated and dry *ELIP* transcripts are accumulated in response to dehydration. To gain further insights into *ELIP* gene function in the moss *S. caninervis*, two *ELIP* cDNAs cloned from *S. caninervis*, *ScELIP1* and *ScELIP2* and both sequences were used as the basis of a transcript abundance assessment in plants exposed to high-light, UV-A, UV-B, red-light, and blue-light. *ScELIPs* were expressed separately in an *Arabidopsis*
*ELIP* mutant *Atelip*. Transcript abundance for *ScELIPs* in gametophytes respond to each of the light treatments, in similar but not in identical ways. Ectopic expression of either *ScELIPs* protected PSII against photoinhibition and stabilized leaf chlorophyll content and thus partially complementing the loss of *AtELIP2.* Ectopic expression of *ScELIPs* also complements the germination phenotype of the mutant and improves protection of the photosynthetic apparatus of transgenic *Arabidopsis* from high-light stress. Our study extends knowledge of bryophyte photoprotection and provides further insight into the molecular mechanisms related to the function of ELIPs.

## 1. Introduction

Light is an essential energy source for all plant life and is used to generate a usable carbon-based energy source for growth via photosynthesis, however, when light intensities exceed the plant’s saturated photosynthesis requirements it has the potential to cause damage to the plant [1]. This potential damage is driven by an overproduction of reactive oxygen species (ROS). The oxygen concentration in chloroplasts is governed by the activity of photosystem II (PSII) and the generation of ROS from the oxygen generated by PSII, e.g., H_2_O_2_, superoxide(O_2_^−^), hydroxyl radicals, and ^1^O_2_* in chloroplasts is closely associated with exposure to excess light [2]. High-light intensities lead to an increase in ROS which, if not scavenged, will result in photo-oxidative damage and photoinhibition [3]. Photoinhibition in turn results in the degradation of carotenoids, bleaching of chlorophylls, and an increase in lipid peroxidation driven by the reactive-oxygen derivatives [4]. Photosynthetic organisms must thus find a balance between the requirements for efficient light harvesting and the danger of accumulating light-induced damage when light absorption exceeds the photosynthesis capacity.

Bryophytes, as one of the earliest diverging lineages of the extant land plants, are believed to have faced tremendous challenges in occupying a variety of ecological niches many of which were characterized by water deficits, high-light intensities, and increased exposure to UV radiation. In meeting these challenges bryophytes have evolved an efficient photosynthetic apparatus and photoprotection mechanisms. The drought-tolerant mosses, *Rhytidium rugosum* and *Ceratodon purpureus*, have demonstrably improved photosystem II (PSII) chlorophyll fluorescence quenching when dehydrated that is hypothesized to provide a means to dissipate excess excitation energy as heat [5]. The desert moss, *Pterygoneurum lamellatum*, exhibits damage to the photosynthetic apparatus that directly correlates to the rate of desiccation; F_v_/F_m_values measured for gametophores dried slowly were similar to undried controls but were significantly reduced in plants dried quickly [6]. *Syntrichia caninervis* and *Bryum argenteum,* as predominant moss species of the Gobi Desert [7], rapidly reassemble thylakoid proteins during rehydration after a desiccation event, presumably as a means to recover photosynthesis and growth [8]. These mosses also quickly increase transcript abundance for genes involved in the light reactions, photorespiration, and other photosynthesis-related genes [9,10], which presumably speeds the recovery of photosynthesis and carbon assimilation [11]. In *Syntrichia caninervis,* the increase in transcript abundance for genes involved in photosynthesis accompanies the rapid restoration and reorganization of PSII and increased chlorophyll synthesis [12]. 

To maintain normal function under high-light stress conditions plants induce light-stress proteins, the Early Light-Inducible Proteins (ELIPs), as part of the protective response [13]. ELIPs were initially described as proteins that were transiently expressed during greening of etiolated seedlings [14,15]. ELIPs are members of a closely related group of proteins that share highly conserved regions of homology: The three-helix *ELIPs*, the two-helix *SEPs* (stress-enhanced proteins), and the one-helix proteins (*OHPs*) along with *HLIPs* (high-light induced proteins), and *SCPs* (small Cab-like proteins) [16]. The *Arabidopsis* genome contains two *ELIP* genes, *Elip1* and *Elip2*, both of which, along with light-induced greening, have been implicated in the control of seed germination, particularly when germination occurs under abiotic stress conditions [17]. ELIPs were also induced in mature green plants exposed to light stress in *Arabidopsis* [18] and are thought to play a role in photoprotection by transiently binding free chlorophyll preventing photo oxidative damage and facilitating energy dissipation that protects the PS-reaction center from photoinhibition [19]. 

The environmental responsiveness of ELIPs, as measured by transcript abundance, has been explored in many plants, primarily in angiosperms. *ELIP* cDNA clones have been constructed from RNA isolated from the alga *D. salina* [20], the bryophytes *S. ruralis* [21], and *Physcomitrella patens* [22], as well as a number of tracheophytes, including pea [23,24], barley [25], *Arabidopsis* [26], wheat [27], and the resurrection plant, *Craterostigma plantagineum* [28]. In addition to cases mentioned earlier, high-light stress induced accumulation of *ELIP* transcripts has been reported for pea [23,24], wheat [27], barley [25,29], grapevine [30], and *Arabidopsis* [26]. Low temperature can also induce *ELIP* gene transcript accumulation in pea [24], barely [25], cabbage [31], and *Medicago sativa* [32]. Salt stress can induce *ELIP* gene transcript accumulation in cabbage [31], *Dunaliella salina* [20], and *Syntrichia ruralis* [21]. In *Madicago sativa* [32], *Arabidopsis* [18], and *S. ruralis* [21] the *ELIP* transcript accumulates during dehydration. In all cases, the implication is that ELIPs play a major role in abiotic stress tolerance. This suggested role of ELIPs in abiotic stress tolerance mechanisms was recently strengthened by the observation that tandem proliferation of ELIP genes, coupled with the elevation of ELIP transcript abundance, is a distinctive feature of the genomes of resurrection plants [33].

In this report, we investigated ELIP transcript abundance in *S. caninervis*, a desiccation-tolerant moss, exposed to different light conditions. The aim was to provide further insights into the ability of *ScELIPs* to photo-protect photosynthetic capacity under light-stress conditions and to determine if light quality also impacted *ELIP* transcript abundance. In this study, we also investigated the functional ability of the *ScELIPs* to protect the photosynthetic machinery using an ectopic expression approach. The cDNA sequences were used to generate transgenic *Arabidopsis Atelip* mutants (lacking one of the *AtELIP* genes (*AtELIP2*) constitutively expressing the *ScELIPs* transcripts with the aim of assessing the ability of the *ScELIPs* to complement the *Atelip* mutant phenotype under different light treatments in different growth phases.

## 2. Results

### 2.1. Multiple Sequence Alignment and Phylogenetic Analysis

We identified 26 Unigenes as *ScELIPs* in the reference transcriptome [10], 6of which have complete open reading frames (ORFs): Unigenes 13021, 40120, 40121, 68225, 16576, and 8044. Unigene 16576 and 8044 each contain 94 predicted amino acids that were of short length to known ELIPs (Appendix A). The predicted pI of Unigene 13021 was 5.58 which was significantly lower than other ScELIPs Unigenes. Unigenes 40121 and 68225 each contain ORFs that were of comparable length to known ELIPs and encode a putative primary amino acid sequence that matched known ELIP proteins. A multiple alignment of ScELIP with other ELIPs is shown in Appendix A, *ScELIPs* have two conserved ELIP domains and exhibit conserved primary sequence regions to known ELIPs [34]. Unigenes 40121 and 68225 were chosen for generation of the target ELIP transgenes and were designated *ScELIP1* and *ScELIP2*. *ScELIP1* (GeneBank number KM363766) was 995-base pair (bp) in length and contained a 711 bp ORF encoding a 236 amino acid putative polypeptide. *ScELIP2* (GeneBank number KM363767) was 904 bp in length and contained a 624 bp ORF encoding a 207 amino acid putative polypeptide as shown in Table 1. 

*ScELIPs* have one Chloroa_b-bind conserved domain like other known ELIPs (Figure 1). The Neighbor-Joining cladogram analysis (Figure 1) clustered *ScELIP1* with other mosses *ELIPs*, such as *SrELIPs* (*Syntrichia ruralis*), *PpELIPs* (*Physcomitrella patens*), and *SfELIP1* (*Sphagnum fallax*), liveworts ELIPs (*Marchantia polymorpha*), and ferns ELIPs (*Onoclea sensibilis*). *ScELIP2,* however, clustered with monocots and eudicots ELIPs, such as *AtELIP1* and *2* (*Arabidopsis thaliana*), *ZmELIP* (*Zea mays*), *OsELIP* (*Oryza sativa*), *GrELIP* (*Gossypium raimondii*), and *BrELIP* (*Brassica rapa*).

### 2.2. Gametophytic Expression of ScELIPs in Response to Light Treatments

To elucidate whether a correlation exists between light quality and/or intensity with *ScELIPs* transcript abundance, the level of *ScELIP1* and *ScELIP2* transcripts were quantitatively assessed in total RNA from gametophytes of *S.caninervis* following exposure to various light regimes using RT-qPCR (Figure 2). *ScELIP1* transcript abundance rapidly declines when the gametophytes are exposed to either UVA or UVB (Figure 2a,b) and does not recover from continued exposure to UVA but exhibits an ability to recover to approximately 50% of control abundance when exposed to UVB for 2 h but not after 4 h or 12 h. *ScELIP2* transcript abundance does not decline in response to exposure to either UVA or UVB but exhibits a peak increase in abundance at 4 h of exposure to UVA and after 2 h of exposure to UVB. In both treatments the level of abundance in the transcripts return to control levels at longer exposure times.

*ScELIP1* transcript abundance increased 7-fold after 4 h of exposure to red-light (Figure 2c) and to similar levels in response to blue-light exposure after 4 h (Figure 2d). *ScELIP2* transcript abundance only increased slightly in response to red-light and only after 4 h of exposure (Figure 2c), but the transcript abundance steadily increased in response to prolonged exposure to blue-light (Figure 2d).

*ScELIP1* transcript abundance increased approximately 5-fold after exposure to red-light for 2 h followed by blue-light for 2 h indicating that blue-light only dampened the response of *ScELIP1* to red-light (Figure 2c). When the treatment was reversed, *ScELIP1* transcript abundance was unaltered from control levels indicating that red-light prevented the transcript abundance response to blue-light (Figure 2d). *ScELIP2* transcript abundance, in response to the identical dual light treatments, was elevated 15-fold with a 2 h exposure to red-light followed by 2 h of blue-light, but exhibited no response when 2 h of blue-light was followed by 2 h of red-light. Thus indicating, as for *ScELIP1* transcript abundance, red-light inhibited the blue-light response.

The transcript abundance for both *ScELIPs* increase with increasing white light intensity. *ScELIP1* transcript abundance increases linearly with increasing light intensity above 60 μmol/m^2^/s to a maximum abundance 9-fold higher than the dark control at 1500 μmol/m^2^/s (Figure 2e). *ScELIP2* transcript abundance increases to a maximum level at 8-fold higher than the dark control at 750 μmol/m^2^/s but declined significantly when light intensity reached 1500 μmol/m^2^/s (Figure 2e). 

### 2.3. Transgenic Lines

We obtained 15 *ScELIP1* and 22 *ScELIP2* transgenic T1 lines following the initial hygromycin screen of T0 transformants (Appendix A). RT-PCR and quantitative real time PCR validated transcript abundance of transgenic lines of *ScELIPs* (Appendix A), the transcript abundance of 35S-*ScELIP1* and 35S-*ScELIP2* transgenic lines were 5-fold to 23-fold higher than the *Atelip* mutant. In normal condition, there isn’t’t visible phenotype difference between the Col-0 WT, the *Atelip* mutant, and the *ScELIPs* transgenic lines. Finally, we chose three transgenic lines of each gene construct that had significant expression levels, and named as *ScELIP1*-line 1, line 2, line 3, and *ScELIP2*-line 1, line 2, line 3.

### 2.4. Germination and Early Seedling Growth under Different Light Treatments

To assess the effect of the expression of *ScELIP* constructs on seed germination under different light stress conditions, germination performance and early seedling morphology was assessed over a seven-day period for mature seeds of the Col-0 WT, the *Atelip* mutant, and two of each of the 35S-*ScELIP1* and 35S-*ScELIP2* transgenic *Atelip* lines (Appendix A and Figure 3). The Col-0 WT, *Atelip* mutant, and the two transgenic had no significant difference. The Col-0 WT and the two transgenic lines had significantly higher germination rates than the *Atelip* mutant until day 4 when red-light (Figure 3c,d) or blue-lightblue-light (Figure 3e,f) was used as the single light source. Germination under UVB light, the Col-0 WT had a significantly higher germination rate than either the *Atelip* mutant or the 35S-*ScELIP* transgenic lines, which had almost identical germination rates, up to 4 days after imbibition (Figure 3g,h). High-light irradiation (1500 μmol/m^2^/s) resulted in a one-day delay of germination for all genotypes (Figure 3i,j). Under high-light, the germination rate for the Col-0 WT was faster than either the *Atelip* mutant or the 35S-*ScELIP* transgenic lines up to day 5, the germination rates for the 35S-*ScELIP* transgenic lines, however, were significantly faster than the *Atelip* mutant up to day 4 (Figure 3i,j). Expression of *ScELIPs* in the *Atelip* mutant increase germination rates but the rates do not attain WT rates when grown in red, blue, UVB, and high-light conditions. 

Exposure to light from each of the sole light sources altered the initial seedling morphology in all genotypes: Red-light severely reduced the size of seedlings and caused hypocotyls elongation, blue-lightblue-light enlarged cotyledons and extended hypocotyls, UVB light resulted in failure of the cotyledons to open and a shortened hypocotyl, and high-light resulted in reduced hypocotyl extension as shown in Appendix A.

### 2.5. Light Treatment Effects on 7-Day Old Seedling Growth and Morphology

Seven-day old seedlings of the Col-0 WT, *Atelip* mutant, and *35S-ScELIP1* and *35S-ScELIP2* transgenic lines were exposed to 4 different light treatments; control conditions, red-light only, blue-light only, and UVB light for seven days and the resulting phenotypes were recorded (Appendix A) and growth parameters quantified (Figure 4). Under normal conditions, the Col-0 WT, *Atelip,* and transgenic lines showed no difference in seedling morphology (Appendix A). Under red-light alone, the number of leaves (Figure 4c) and overall root lengths (Figure 4a) were reduced in comparison to the controls and equally for all of the genotypes, only root number (Figure 4b) was unaffected by the red-light treatment. The red-light treatment also caused etiolation of the leaves (Appendix A) and reduced the chlorophyll content (Figure 4d). Under blue-light alone, the overall root length was significantly reduced for all genotypes compared to the control (Figure 4a). Conversely, the seven- day exposure to blue-light resulted in an approximate ten-fold increase in secondary root and the increase was significantly greater for the *Atelip* mutant and the *35S-ScELIP* transgenics (Appendix A and Figure 4b). Exposure to blue-light also resulted in chlorosis of the leaves of the Col-0 WT with a concurrent reduction in chlorophyll, however, the *Atelip* mutant and the *35S-ScELIP* transgenics remained green and the chlorophyll content was only reduced to approximately 50% of the controls (Figure 4d). The UVB treatments inhibited seedlings growth in all genotypes with large reductions in root length and leaf number (Appendix A and Figure 4a,c). The leaf chlorophyll contents also declined when compared to controls but the seedlings remained green for all genotypes (Figure 4d).

### 2.6. Seedling Growth of WT and Transgenic Lines under High-Light Conditions

Seven-day old seedlings of the Col-0 WT, *Atelip* mutant, and transgenic lines were exposed to high-light stress (1500 μmol/m^2^/s), and morphology and the chlorophyll contents were monitored after 2 h exposure and 24 h recovery 22 ± 2 °C with a 14 h light/10 h dark cycle, and 60–75% relative humidity, and light at PPFD of 100 μmol/m^2^/s (Figure 5). Following the high-light exposure, the transgenic lines appeared greener than the Col-0 WT and *Atelip* (Figure 5a,b) which was reflected in the chlorophyll contents that were significantly increased (Figure 5c). Corresponding with chlorophyll contents, carotenoid contents of the transgenic lines were significantly higher than the Col-0 WT and *Atelip* mutant (Figure 5d).

### 2.7. The Effect of High-Light Stress on Arabidopsis Plantlets

Four-week old *Arabidopsis* plantlets, including WT, *Atelip* mutant, and two transgenic lines of *ScELIP1* and *ScELIP2*, were exposed to high-light stress (1000 μmol/m^2^/s) for 3 days and continuously for 3 weeks. The effect of high light appears to be more severe in the *Atelip* mutant line after the 3-day treatment than the Col-0 WT and all of the *35S-ScELIP* transgenic lines (Figure 6). All genotypes were severely affected by 3 weeks of exposure to high-light, although the *Atelip* mutant does appear to be more severely damaged. 

The chlorophyll fluorescence parameters measured were consistent with visual estimates of leaf damage. The 3-day high-light treatment resulted in a significant decline in the F_v_/F_m_ values (Figure 7) indicative of an inhibition photosynthesis under high-light stress. After 3 days of high-light treatment, the F_v_/F_m_ values were so low as to be unreliable and/or undetectable so this data was not added to the analysis. The chlorophyll fluorescent parameters decreased as the stress time extend in all *Arabidopsis*, it indicated that high-light increased the decline of parameters, the PSII complex have been photo-inactivated, and chlorophyll began to degrade (Figure 6 and Appendix A), a severely compromised photosynthetic apparatus. However, the *35S-ScELIPs* lines exhibit a higher retention of the F_v_/F_m_ (Figure 7), indicating an increase in protection from photo-induced inactivation and thus potentially overcoming the lack of the *ELIP2* gene product that is missing in the *Atelip* mutant.

Pigment contents were measured prior to the treatment (time zero), after 3 days and after 3 weeks of the high-light treatment. Both chlorophyll a and b and carotenoid contents decreased in both the 3-day and 3-week treatments with high-light (Appendix A). Although, the pigment content of the *35S-ScELIPs* lines were not elevated with respect to the Col-0 WT controls they were significantly greater than the levels measured in the *Atelip* mutant line, indicating that the *35S-ScELIPs* constructs can rescue the *Atelip* mutant with regards to the effect of a lack of ELIP2 on high-light induced photodamage and pigment loss. The apparent rescue of the effect of the *Atelip* mutation on photosynthetic capacity is also reflected in the overall restoration of the Chla/Chlb and Car/Chl ratio of all *35S-ScELIPs* (Appendix A). 

Transcript abundance was assessed for *ScELIP1* and *ScELIP2* and other photosynthesis related genes, in all genotypes, before treatment (0 h), after 2 h, 3 days, and 3 weeks exposure to high-light stress using RT-qPCR (Figure 8). *ScELIP1* and *ScELIP2* exhibit moderate levels in the controls (0 h), presumably indicating the level of expression obtained with the *35S* promoter. *ScELIP1* transcript abundance rapidly increased 35–70-fold in the *35S-ScELIP1* lines following exposure to high-light for 2 h but subsequently declines to low abundance after 3 days and negligible levels after 3 weeks of exposure to high-light (Figure 8a). *ScELIP2* transcripts rapidly declined in abundance upon exposure of the *35S-ScELIP2* transgenic to high-light, however, *ScELIP2* transcript abundance significantly increased to 49–69-fold in response to prolonged exposure to high-light for 3 weeks (Figure 8b). *ScELIP2* transcripts did not accumulate to significant levels in the Col-0 WT even after exposure to high-light for 3 weeks. 

Transcript abundance for 3 of the AtELIP family of related proteins were also altered in response to high-light. The abundance of *OHP* transcripts peaked at 3 days of exposure to high-light in Col-0 WT and also in the *35S-ScELIPs* lines but to a lesser extent. The OHP transcripts in the *35S-ScELIPs* lines were significantly more abundant than the same transcript in the *Atelip* mutant. In all cases OHP transcripts declined in abundance at 3 weeks (Figure 8d). *OHP2* transcripts accumulate to levels 5–15-fold greater than control levels (0 h) in the Col-0 WT and *35S-ScELIPs* lines after 3 weeks of exposure to high-light and to levels significantly greater than in the *Atelip* mutant (Figure 8e). *SEP2* transcripts increased in abundance after exposure to high-light for 2 h in the Col-0 WT and *35S-ScELIPs* lines, in both cases to higher levels than in the *Atelip* mutant, declined after 3 days of high-light exposure and then returned to near control levels after 3 weeks (Figure 8f). 

*LHCA2* transcripts declined in abundance response to exposure to high-light at 2 h and 3 days, and slightly increased at 3 weeks in the *35S-ScELIP* lines. However, in the Col-0 WT LHCA2 transcripts increased in abundance after 3 days of high-light exposure and remained high throughout the length of the treatment (Figure 8g). *LHCB4.2* transcript abundance in the *Atelip* mutant, remained constant throughout the 3-week exposure to high-light. In the Col-0 WT and both transgenic lines, *LHCB4.2* transcript abundance declined during the first 3 days of exposure to high-light but was dramatically elevated in abundance in the samples exposed for 3 weeks (Figure 8h).

In the Col-0 WT, *PSBD* transcript abundance was elevated 5-fold after 3 days of exposure and remained elevated throughout the high-light treatment but in the *35S-ScELIP* transgenic lines *PSBD* transcripts only accumulated to elevated levels in the 3-week exposure samples. The *PSBD* transcript abundance in the *Atelip* mutant remained at relatively low levels throughout the treatment (Figure 8i). *PSBS* transcript abundance initially high in the control leaves of the *35S-ScELIP1* lines declined in response to exposure to high-light with a moderate recovery at 3 weeks. *PSBS* transcript abundance in the control Col-0 WT exhibited a similar response to that of the *35S-ScELIP1* lines but made a full recovery at 3 weeks. In the *35S-ScELIP2* lines, *PSBS* transcript abundance maintained initial levels through the first 3 days of exposure to highlight but were significantly elevated above control levels at 3 weeks (Figure 8c).

## 3. Discussion

In this study, we identified and characterized two *ELIP* genes from the *S. caninervis* transcriptome. *ScELIP1* has close primary amino acid sequence similarity to the *ELIPs* of the non-vascular plants *Physcomitrella patens*, *Sphagnum fallax,* and *Syntrichia ruralis* [21] excepting one kind of fern (*Onoclea sensibilis*). *ScELIP2* that is more similar to the *ELIPs* of vascular plants such as *Arabidopsis* [26], *Zea mays* [35], etc. (Figure 1). According to the amino acid sequence similarity of *ELIPs*, bryophytes ELIPs were clearly separated from monocots and eudicots *ELIPs*, because bryophytes are a special evolutionary group. Neither of the *ScELIP* sequences fully corresponds to either of the two *Arabidopsis ELIP* genes which, by ectopically expressing the *ScELIP* genes independently in the *Arabidopsis Atelip* mutant background, allowed us to evaluate both the possible activity of the *ScELIP* transcripts in photoprotection of chloroplasts in situ but also to determine if the moss ELIPs could rescue the *Atelip* phenotype resulting from the loss of *AtELIP2.*

In *Syntrichia caninervis ScELIP1* and *ScELIP2* transcripts abundance were elevated by different light stresses (Figure 2). *ScELIP1* and *ScELIP2* transcripts exhibit similar responses, both exhibit greater accumulation in response to blue and high-light exposure and both exhibit transcript accumulation only after a 4 h exposure to red-light, more evident for *ScELIP1*. Our understanding of how or why *ELIPs* respond to different light qualities is still very limited, but in pea seedlings *ELIPs* transcripts are specifically induced by blue-light but not by red-light, and the accumulation of blue-light induced *ScELIPs* transcripts are significantly repressed by low intensity red-light [23]. These findings are consistent with the *S. caninervis* data presented here with the exception that prolonged exposure to red-light alone does result in an elevation *ScELIP* transcript abundance. The two *Arabidopsis ELIP* genes can be induced in dark grown *Arabidopsis* exposed to illumination by high intensities of both red and blue-light [18] which again are consistent with observations. In *Arabidopsis*, *ELIP* transcript abundance was elevated by UVA but not by UVB [36]. In *S. caninervis*, *ScELIP1* transcripts declined in abundance in response to UVA and *ScELIP2* transcript levels only exhibited a response to UVA after a 4 h exposure (Figure 2a). UVB exposure caused a general decline in *ScELIP1* transcripts but generated an increase in abundance of ScELIP2 transcripts after 2 h of exposure. Clearly, the *S. caninervis* ELIP response to UV light differs from that seen in *Arabidopsis*, perhaps indicating a more complex role in UV protection mechanisms in the moss. However, this possibility must be tempered by differences in experimental protocols and by the distinct possibility that the moss genome contains more than two ELIP genes as would be suggested by the observation that the genomes of desiccation tolerant plants are rich in ELIP genes [33].

It is clear from our analyses that ELIP transcript abundance in *S. caninervis* is altered by light quality (UVA and B, red and blue-light), suggesting that the promoters that control the transcription of *ELIP* genes contain elements that respond to light signal transduction pathways. At the present time, the promoter sequences for the *S. caninervis* ELIP genes are unavailable but ELIP promoters from other species have been investigated. In *Ginkgo biloba*, high-light defense and stress responsive elements were delineated in the promoter region of *GbELIP* [37]. In *Arabidopsis*, *ELIP1* expression is enhanced by red, far-red, and blue-light and that phytochrome A and B are involved in the signaling pathway along with the transcription factor HY5 [18]. The *Arabidopsis ELIP2* promoter region was later demonstrated to contain two regulatory elements, that form a regulatory unit, that regulate *ELIP2* expression in response to UVB, high-light, and cold stress [38]. If similar elements appear, the *ScELIP* promoter might add to our understanding of the evolution of light response transduction pathways in plants. 

The significant and rapid elevation in the *ScELIPs* transcript abundances when gametophytes are transferred from the dark to high-light is consistent with similar observations in *Arabidopsis* [39], *Hordeum vulgare* [25], *Pisum sativum* [23], *D. salina* [20], and *Syntrichia ruralis* [21] and supportive of the hypothesis that ELIPs are useful indicators of high-light stress and the initiation of photoprotection mechanisms during the early phases of photodamage [40]. 

Light stress has been demonstrated to exert influence on seed germination and photomorphogenesis during the initial developmental stages of seedling growth [17]. Our results supported the observation that light quality alters photomorphogenesis following germination (Appendix A) such that the hypocotyl was shortened and the cytoledons remained closed following germination under UVB light [41]. In a recent report, UVB photoreceptor UVR8 (UV resistance locus 8) promotes rapid PIF (phytochrome interacting factors) degradation and reduces PIF abundance, which lead to UVB inhibits shade avoidance, finally, UVB inhibits stem elongation of Sharma et al. [42]. While this phenotype was not altered by the expression of the *35S-ScELIPs* in the transgenic seeds. The addition of the *35S-ScELIPs* genes into the *Atelip* mutant line significantly improved the germination rate above that of the mutant alone without altering leaf number, root number, or root length with the exception of an increase in root number in the *35S-ScELIP* transgenics under blue-light (Figure 4). 

The complementation of the *35S-ScELIP* genes to the *Atelip* mutant line enhanced the survival rate of seedlings under high-light stress in part by minimizing chlorophyll degradation. The transgenic lines, by protecting the chlorophyll, were greener in appearance than even the Col-0 WT lines after 24 h of recovery following exposure to high-light conditions and also had a higher percent survival (Figure 5). This was also evident in the more mature plant, where the transgenic lines exhibited significantly less photodamage under prolonged exposure to high-light intensities than the mutant line (Figure 6). The improved protection to photodamage observed in the *35S-ScELIP* transgenics was reflected in their improved photosynthetic performance measures compared to the *Atelip* mutant increased F_v_/F_m_ (Figure 7). These data suggest that the *ScELIPs* can protect PSII functionality of *Arabidopsis* plantlets under high-light conditions. ELIP accumulation under high-light stress has been correlated with photoinhibition of PS, degradation of the D1 protein, and changes in the level of pigments in the mature plant [36,43]. These conclusions are consistent with our data indicating enhanced PSII protection in the *35S-ScELIP* transgenics. 

The results of this study indicate that the physiological function of ELIPs are related to the protection of chlorophyll, and thus enhance the protection of the chloroplasts from photooxidative stress. The *35S-ScELIP1* and *35S-ScELIP2* lines also have improved carotenoid levels than those recorded for the *Atelip* mutant following high-light exposure and thus it is possible that *ScELIPs* may also play a role in stabilizing the photosynthetic apparatus and is consistent with the idea that ELIP acts as a photoprotectant for PSII in the thylakoid membrane and as such would play a major role in the ability of *Syntrichia caninervis* to survive in extreme conditions, as suggested by VanBuren et al. [33], and that it regularly encounters in the Gobi Desert.

The *35S-ScELIP1* and *35S-ScELIP2* lines have a significant response to high-light (Figure 8a,b) even though their transcription is controlled by the *35S* promoter. It is possible that the overall transcription rates affected by high-light or transcript abundance under high-light is partially controlled by transcript stability. The former seems unlikely given the variability in transcript abundance for other genes in response to high-light (Figure 8). It is possible that transcript abundance was controlled in part by the altered stability but this possibility was not investigated. It is also possible that the 35S promoter used in these constructs is affected by exposure to high-light but there are no reports of this as a possible explanation. It is intriguing that the transcript abundance profile for *ScELIP1* is more similar to the native *Arabidopsis SEP2* and the transcript abundance profile for *ScELIP2* is more similar to *OHP2* than to other photosynthetic related transcripts. If the 35S promoter, which is active in the untreated controls, is not high-light inducible then it could suggest the presence of a post-transcriptional mechanism that may control the abundance of the transcripts of three *ELIP* families of protective proteins. The transcript abundance profiles in response to both short and long exposure to high-light for all of the photosynthetic related transcripts in the genotypes we investigated were complex. However, it was evident that the *Atelip* mutant was, with the exception of the *SEP2* transcript which was elevated above control levels (Figure 8f), deficient in proteins that function to protect PSII. The ectopic expression of the *ScELIP* transcripts in the *Atelip* mutant appears to provide the necessary protective proteins (at least the transcripts) which overcomes the high-light phenotype of the mutant. In addition, the transcript abundance profiles for all genes and genotypes indicate either a recovery of or elevation in transcript abundance after 3 weeks of exposure to high-light which may be indicative of the ability of *Arabidopsis* to acclimate to this environmental stress.

## 4. Materials and Methods

### 4.1. Plant Material and Growth Conditions

Dry *Syntrichia caninervis* gametophytes were collected from the Gurbantunggut Desert in the Xinjiang Uygur Autonomous Region, China (44° 32′ 30″ N, 88° 6′ 42″ E) and kept in the dark in a paper sack at room temperature. Dry gametophytes were fully hydrated on filter paper saturated with MINIQ-filtered water (8 mL) in glass petri dishes for 24 h at 25 °C, with light at a photosynthetic photon flux density (PPFD) of 50 μmol/m^2^/s, prior to the described light treatments.

Seeds of *Arabidopsis thaliana* L. cv. Columbia and the *Arabidopsis* mutant *Atelip* were germinated on MS medium [44] after surface sterilization with 70% ethanol for 30 s followed by 3% sodium hypochlorite for 7 min and washed 5 times with sterile water. Seedlings were transferred to 5 cm diameter pots at 4–6 leaf stage containing autoclaved peat substrate (Pindstrup, Mosebrug) and grown under controlled conditions at 22 ± 2 °C with a 14 h light/10 h dark cycle, 60–75% relative humidity, and light at PPFD of 100 μmol/m^2^/s. *A. thaliana* (Col-0) and the mutant *Atelip* (NASC ID: N544166)*,* a T-DNA insertion mutant in the promoter region of *ELIP2* [45], were obtained from The European Arabidopsis Stock Centre.

### 4.2. Bioinformatic Analysis of Syntrichia Caninervis ELIP Sequences

Publicly available *ELIP* sequences of *A. thaliana* and *Physcomitrella patens* were used to screen the transcriptome of *S. caninervis* [10] for single copy transcripts (designated as unigenes) that were putative *ELIP* coding sequences. We translated each unigene to obtain the amino acid sequence using ORF finder. All generated amino acid sequences were investigated using NCBI-blastp. The final sequences used to construct the *ScELIP* clones were consistent with annotated *ELIP* genes in the NCBI database. A molecular phylogeny was constructed using MEGA7.0 with the neighbor-joining method and 500 bootstrap replicates [46] utilizing the eudicots, monocots, ferns, and bryophytes ELIP sequences obtained from public resources and presented in additional data (Appendix A). All of ELIP genes domain prediction was conducted with the Pfam database (http://pfam.xfam.org/search). Further physiological and biochemical properties prediction for two ScELIPs was obtained on ExPASy (https://www.expasy.org/).

### 4.3. Extraction of RNA and RT-qPCR Based Expression Analysis

Total RNA was isolated from 0.1 g of *S. caninervis* gametophytes using the TRIzol reagent (Qiagen, Berlin, Germany) according to the manufacturer’s protocol. The cDNA was synthesized from 1 μg of total RNA using random hexamer primers with the PrimeScript RT reagent kit including gDNA Eraser (Real time; Takara, Japan) according to the manufacturer’s protocol and stored at -20 °C until use. The cDNA was diluted 10-fold with H_2_O to serve as a template for Real-time PCR. Each 20 μL real-time PCR reaction contained 2 μL cDNA, 0.5 μL each of forward and reverse primers as indicated below (10 μM for each primer), and 10 μL of SYBR Premix Ex Taq^TM^ (Takara Biomedical Technology, Takara, Japan). Real-time quantitative PCR was performed in 96-well plates with the CFX96 Real-Time PCR Detection System (Bio-Rad, New York, NY, USA) according to the manufacturer’s instructions [47], with three technical replicates and two biological replicates. The RT-qPCR protocol was as follows: 30 s initial denaturation at 95 °C, 40 cycles of 94 °C for 5 s, and 60 °C for 30 s. Parallel reactions to amplify *α-TUB2* were used to normalize the amount of template. The forward and reverse primers used for *ScELIP1* were 5′ GGCTATGCTGGGATTCGTGT 3′ and 5′ TTGATCTGGTTCCTGGCGTC 3′, the primers for *ScELIP2* were 5′GCAATGATCGGACTTGTCGC3′ and 5′ ACCTGCTGTGAACAATCCGT 3′. All PCR efficiencies were >95%. Transcript abundance data were collected using Bio-Rad CFX Manager.

Hydrated *S. caninervis* gametophytes were exposed to the following environmental treatments: A progressive series of PPFD, Ultraviolet A (UVA), Ultraviolet B (UVB), red-light, and blue-light. Fully hydrated *S. caninervis* gametophytes were transferred to MINIQ-filtered water saturated filter paper in clean petri dishes for all treatments. The light series treatment consisted of a 2 h exposure to either 0 μmol/m^2^/s, 60 μmol/m^2^/s, 225 μmol/m^2^/s, 750μmol/m^2^/s, or 1500 μmol/m^2^/s provided by a cool-white fluorescent light lamp (Philips lighting). For the ultraviolet radiation treatment gametophytes were exposed to either UVA (380 nm) or UVB (308 nm) irradiation for 0 h, 1 h, 2 h, 4 h, 12 h under a fluorescent light lamp (Philips lighting) that delivered a PPFD of 10 μmol/m^2^/s. In the red-light treatment, gametophytes were either exposed to red-light using a monochromatic light-emitting diode (LED, Philips lighting, Amsterdam, Netherlands) at 655 nm, and PPFD of 100 μmol/m^2^/s for 0, 2, 4 h, or exposed to red-light for 2 h and then transferred to blue-light for 2 h. In the blue-light treatment gametophytes were either exposed to blue-light using a monochromatic light-emitting diode (LED, Philips lighting) at 450 nm and PPFD of 100 μmol/m^2^/s for 0 h, 2 h, 4 h, or exposed to blue-light for 2 h and then transferred to red-light for 2 h. Each treatment consisted of three biological replicates. 

### 4.4. Cloning of ScELIP Genes and Establishment of Transgenic Lines

The transcripts *ScELIP1* (Unigene 40121) and *ScELIP2* (Unigene 68225) were chosen from the *Syntrichia caninervis* transcriptome database [10]. The full-length coding sequence for each gene product was obtained by RT-PCR using the cDNA previously constructed for the RT-qPCR based expression analysis. The RT-PCR was performed using gene-specific primers: For *ScELIP1* the forward and reverse primers were 5′ GGGGTACCC ATGGCAGCGATGGCG 3′ and 5′ CGAGATCTCGCTAGACAGGGAAGCG 3′, and these primers include inserted *KpnI* and *XbalI* enzyme sequences. For *ScELIP2,* the forward and reverse primers were 5′ CGGGATCCATGGCGATGACTTT 3′ and 5′ CGAGATCTCGTTATACAAGTGGGC 3′, and these primers include *Bam HI* and *XbalI* site sequences. The PCR product was extracted from agarose gel electrophoresis (0.8% agarose, 5 v/cm voltage for 20 min) using the Gel Extraction Kit (OMEGA, St. Louis, MI, USA), and 4 μL PCR extract production, 1 μL pMD 19-T vector, 1 μL T4 DNA ligase, 2 μL 10× ligation buffer, and 12 μL H_2_O for 2 h (TaKaRa, Japan), linking the product with pMD 19-T vector. The sequences were validated by sanger sequencing at the Beijing Genomics Institute (BGI) to ensure that there were no PCR introduced mutations into the coding sequences.

### 4.5. Transformation of A. Thaliana

The *ScELIP1* ORF was removed from *ScELIP1*-19T by digestion with *KpnI* and *XbalI* and cloned into *KpnI* and *XbalI*—digested pCAMBIA 1301 to construct a 35S-*ScELIP1* gene (Appendix A). Likewise, the *ScELIP2 ORF* was removed from *ScELIP1*-19T by digestion with *Bam HI* and *XbalI* and cloned into *BamHI* and *XbalI*—digested pCAMBIA 1301 to construct a 35S-*ScELIP2* gene (Appendix A). The p*ScELIP1*-CAMBIA 1301 and p*ScELIP2*-CAMBIA 1301 were introduced separately into an *Agrobacterium tumefaciens* strain EHA105 by triparental mating as described by Hoekema et al. [48] and introduced into 6-week old *Arabidopsis ELIP* mutant *Atelip* plants by the floral dip method [49]. T1 transgenic plants were selected by germination on MS agar plates containing Hygromycin (80 mg/L). After 7 days, the majority of the seedlings turn yellow or white and die. After 10 days, those seedlings that continued to stay green and grow were considered as stable transformants. The stable transformants were verified by PCR by using the same gene-specific primers used to isolate the coding sequences for each *ScELIP*. Primary transformants were shelfed to generate T2 homozygous transgenic seed stocks for production of T3 homozygous plants used in the phenotypic analysis. 

### 4.6. Seed Germination Assay

Petri plates containing 1/2 MS agar media (1%) were divided into four quadrants and approximately 50 seeds were sown of each genotype into separate quadrants: *Arabidopsis* Col-0 wildtype (WT), the *Atelip* mutant, and two transgenic *Arabidopsis* lines of *ScELIP1* or *ScELIP2*. Seed germination, when the radicle protruded through the seed coat, was scored from the first day after sowing. The seed germination assay was conducted under standard conditions (22 ± 2 °C, 14 h light/ 10 h dark cycle, 60–75% relative humidity, and PPFD of 100 μmol/m^2^/s) as a control or under the following treatments; irradiated by red, blue, UVB, or high-light (1000 μmol/m^2^/s) as their single light source (as mentioned in 2.3 Extraction of RNA and RT-qPCR based expression analysis) at 22 °C 14 h light/10 h dark cycle for 6 days. 

### 4.7. Phenotypic Analysis of Transgenic A. Thaliana

Sterile WT, *Atelip* mutant, and *ScELIPs* transgenic *Arabidopsis* seeds were vernalized for 2 d and germinated on 1/2 MS agar medium under control conditions of 22 °C and 14 h light/10 h dark cycle and PPFD of 100 μmol/m^2^/s. Seven-day old seedlings, of similar size, were chosen for exposure to experimental treatments. WT, *Atelip* mutant, and transgenic *ScELIPs* lines, were cultured on 1/2 MS medium for 7 days at 22 °C 14 h light/10 h dark cycle. Individual light treatments were administered for 7 days at 22 °C 14 h light/10 h dark cycle and PPFD of 100 μmol/m^2^/s under the appropriate light source of either red, blue, or UVB as described for the gene expression study. After all treatments the number of leaves, roots, and the length of the longest root were recorded.

For the high-light treatment, seven-day-old seedling, grown under control conditions of 22 °C and 14 h light/10 h dark cycle and PPFD of 100 μmol/m^2^/s, were briefly exposed to light at an intensity of 1500 μmol/m^2^/s for 2 h and subsequently transferred to normal conditions for 24 h to recover. Following recovery, the seedlings were assessed for chlorophyll content as described in a later section.

### 4.8. High-Light Treatment of Plantlets

Seedlings of WT, *Atelip* mutant, and *ScELIPs* transgenic lines of *Arabidopsis* were grown in 5 cm diameter pots in a peat substrate (Pindstrup, Mosebrug, Ryomgård, Denmark) and grown under well-watered controlled conditions at 22 ± 2 °C with a 14 h light/ 10 h dark cycle, PPFD of 100 μmol/m^2^/s, and 60–75% relative humidity. After four weeks plants were irradiated with high-light (1000 μmol/m^2^/s) at 22 °C 14 h light/10 h dark cycle under well-watered conditions. F_v_/F_m_ was measured before treatment (0 h), after exposure to high-light for 3 days and then continuously for 3 weeks. Leaf samples were taken for chlorophyll and carotenoid analyses before and after 3 days and 3 weeks of high-light treatment. 

The relative transcript abundance was assessed for *ScELIP1*, *ScELIP2*, One-Helix Protein (*OHP*), *OHP2*, two-helix protein (*SEP2*), PSII CP29 antenna protein (*LHCB4.2*), PSI antenna protein (*LHCA2*), PSII CP22 protein (*PSBS*), and PSII D2 protein (*PSBD*), after exposure to high-light for 0 h and 2 h, 3 days and 3 weeks. Each treatment consisted of 3 biological replicates. The method of extraction of RNA and RT-qPCR based expression analysis was as stated previously. To compare relative expression levels, Col-0 WT and all *ScELIPs* transgenic lines of *Arabidopsis* were compared to the *Atelip* mutant, and *Atα-TUB* was used as a reference gene; primer sequences are listed in Appendix A.

### 4.9. Pigment Analysis

Chlorophyll content was measured according to the methods described by Ritchie 2006 [50]. Leaf samples (50 mg Fwt) were collected from three separate plants for each line and at each treatment phase. Pigments were extracted by incubation of the leaf samples in 2 mL of 96% ethanol (room temperature ~25 °C) for 4 h in darkness with constant agitation. The extracts were centrifuged at 10,000 rpm for 2 min and the supernatants removed for analysis. Supernatants were analyzed spectrophotometrically at wavelengths of 470, 649, and 665 nm using a UV–Visible spectrophotometer (Biomate 3S, Thermo Fisher Scientific, Waltham, MA, USA). The concentration of the chlorophyll a, b, and total carotenoids were determined using the following equations: Chl a = 13.95 × OD_665_ − 6.88 × OD_649_, Chl b = 24.96 × OD_649_ − 7.32 × OD_665_, and Car = (1000 × OD_470_-2.05 × Chl a-114.8 × Chl b)/245. The total pigment content in mg/g =N × C × V/W, “N” represents dilution ratio, “C” represents pigments concentration (mg/mL), “V” represents the volume of extracting solution (mL), “W” represents sample fresh weight (g).

### 4.10. Fluorometric Assessment of Photosynthetic Performance

Photosynthetic performance of *S. caninervis* at a different light intensity and times were assessed by pulse amplitude modulated fluorometry using a portable chlorophyll fluorometer (PAM 2500) (Heinz, Walz, Berlin, Germany). Measurements of chlorophyll fluorescence were recorded in situ on the mid portion of the uppermost fully mature leaf. The saturation pulse method was used to calculate the F_v_/F_m_, the parameter settings were based on the recommendations of Zhang et al. [51]. All parameters were measured on 6 individual plants and the values were averaged for each genotype. 

### 4.11. Statistical Analyses

All statistical analyses were performed using Statistical Product and Service Solutions (SPSS) 16.0 software (SPSS Inc., Chicago, IL, USA). Data were compared using one-way ANOVA, Dunnett’s T3 was used to examine the difference in significance of ANOVA results, and values were considered as significantly statistically different at *p* < 0.05, or distinctly statistically different at *p* < 0.01. Error bars represent standard deviations.

## 5. Conclusions

In *Syntrichia caninervis*, *ScELIP1* and *ScELIP2* transcript abundance were activated by multiple light stresses, especially increasing high-light intensity. Expression of *ScELIPs* in the *Atelip* mutant increase germination rates but the rates do not attain WT rates when grown in red, blue, UVB, and high-light conditions. Different light quality changed morphology after the germination and seedling stage, while this phenotype was not altered between Col-0 WT, *Atelip* mutant, and *35S-ScELIPs* transgenic lines. Two *ScELIPs* can protect chlorophyll and accumulation of photoprotective pigments such as carotenoid under high-light stress. High-light leads to transcriptional suppression of genes encoding antenna proteins such as *LHCA2*, *LHCB4.2*, *PSBS* at 2 h and 3 days, but slightly increases at 3 weeks. High-light causes transcriptional activation of genes encoding proteins responsible for photoprotection, such as *ScELIP1* and *ScELIP2* and 3 of the *AtELIP* family (*OHP*, *OHP2*, *SEP2*). These findings can clarify the process of ELIPs of bryophyte response to different light quality and/or intensity especially to high-light and is important for land plant evolutionary adaptation.

## Figures and Tables

**Figure 1 ijms-21-01411-f001:**
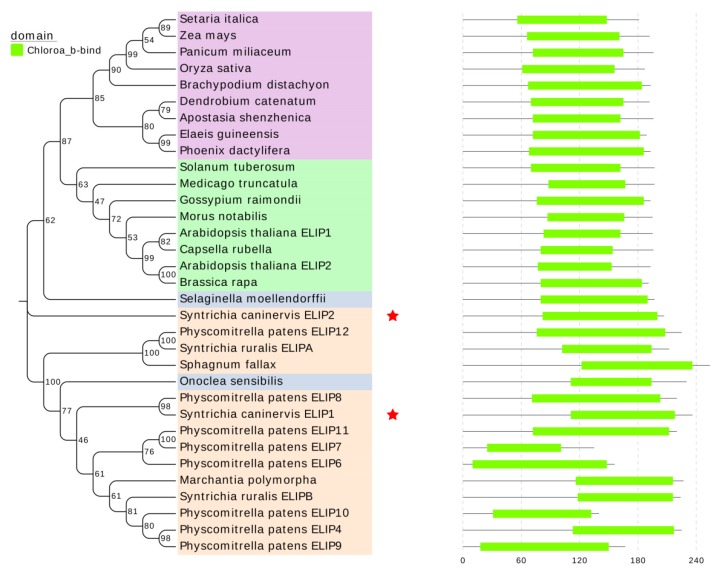
The phylogenetic tree inferred from the deduced protein sequence of ScELIPs and other 34 ELIP amid acid sequences represented by 24 additional taxa and conserved domain. In the left pannel, numbers above the lines represent bootstrap percentages (based on 1000 replicates). A bootstrap value of 100% indicates branches that were supported in all replicates of resampling of data. The scale bar indicates the number changes per unit length. The two ScELIPs were highlighted by a red star. Different colors represent different evolving groups, pink (bryophytes), blue (ferns), purple (monocots), and green (eudicots). In the right pannel, the conserved domain of ELIPs corresponding to the left pannel.

**Figure 2 ijms-21-01411-f002:**
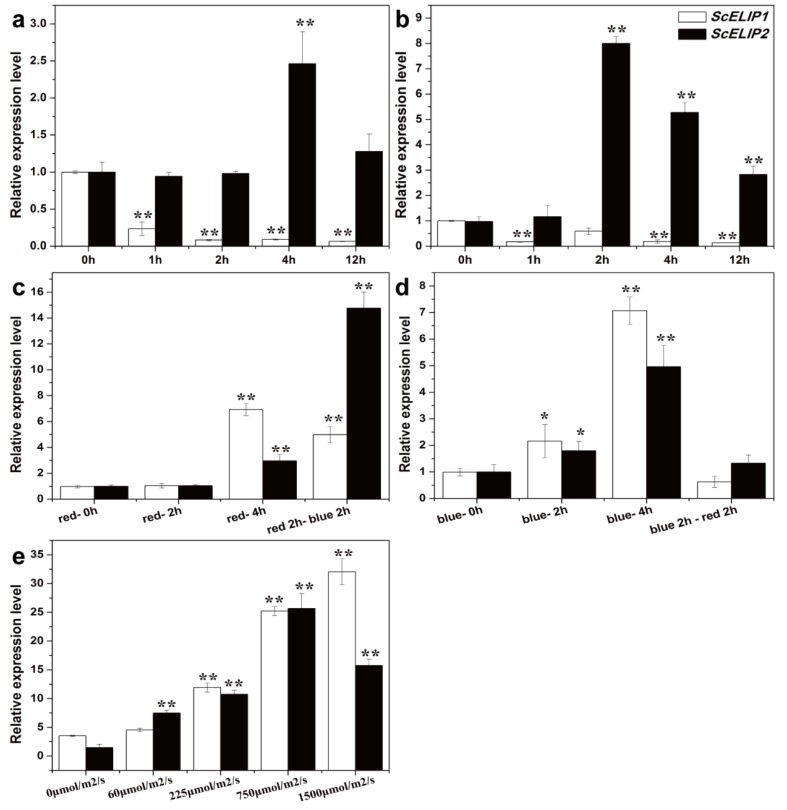
*ScELIP1* and *ScELIP2* transcript relative abundance assessments in *Syntrichia caninervis* gametophores exposed to various light treatments. The mean (each column) and SD (error bar) were calculated with three biological replicates and three technical replicates of each biological replicates. * *p* < 0.05; ** *p* < 0.01. P-values were obtained from Dunnett’s T3 test comparing different time points and treatment with control. Asterisk represents statistically significant difference with 0 h in (**a**–**d**), and 0 μmoL/m^2^/s in (e). ultra violet A (UVA) light treatment (**a**), ultra violet B (UVB) light treatment (**b**), red-light treatment (**c**), blue-light treatment (**d**), high-light treatment (**e**).

**Figure 3 ijms-21-01411-f003:**
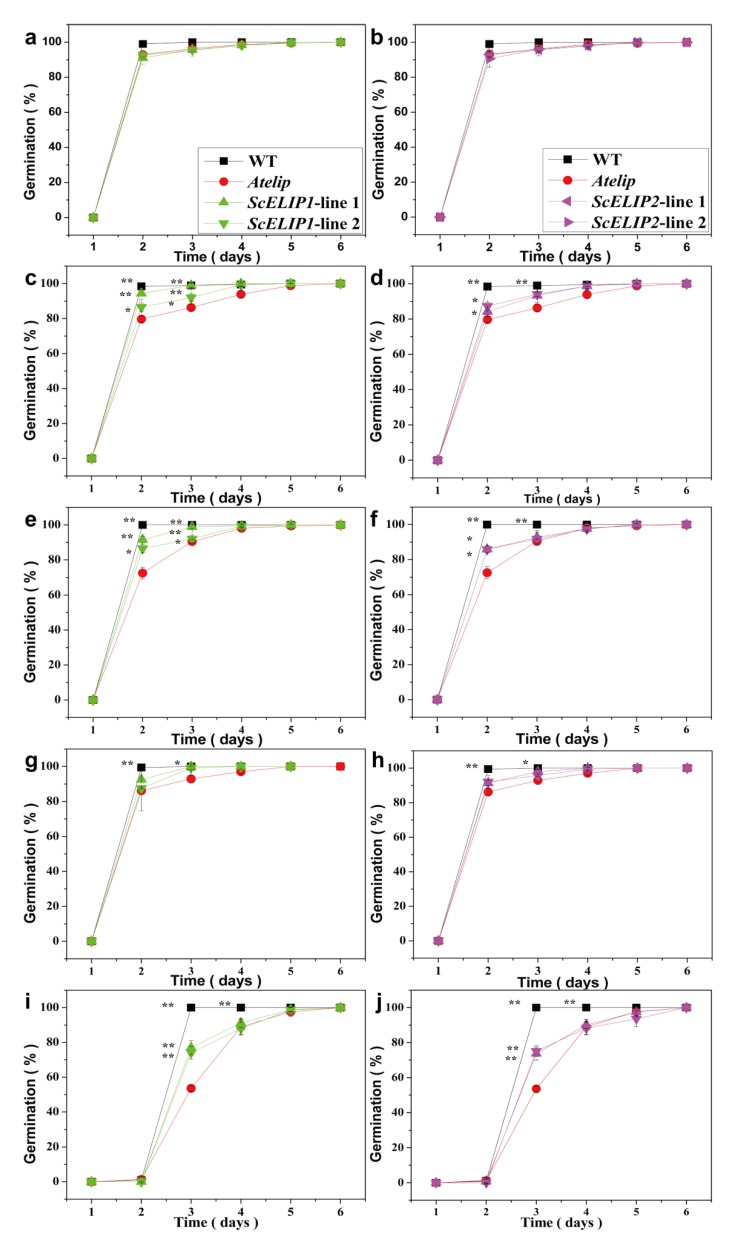
Seed germination rate for Col-0 WT (*Arabidopsis thaliana* L. cv. Columbia), *Atelip* mutant, and *35S-ScELIP* lines under different light treatments. The graphs depict germination rates over six days exposure to the different light regimes: Control condition (**a**,**b**), Red-light alone (**c**,**d**), Blue-light alone (**e**,**f**), UVB light alone (**g**,**h**), High-light irradiation (1500 μmol/m^2^/s, **i**,**j**). Error bars represent the SD (standard deviations) of three biological repeats. * *p* < 0.05; ** *p* < 0.01. P-values were obtained from Dunnett’s T3 test comparing WT and the transgenic lines with the *Atelip* mutant, respectively. First set, *35S-ScELIP1* line 1 and line 2; Second set, *35S-ScELIP2* line 1 and line 2.

**Figure 4 ijms-21-01411-f004:**
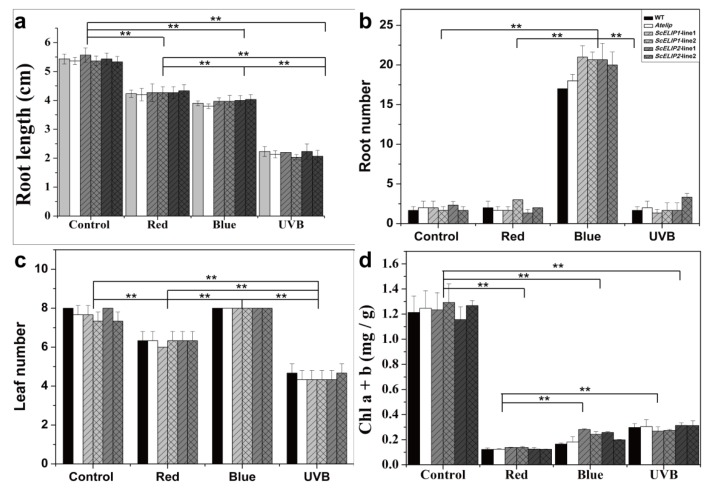
Morphological responses of Col-0 WT, *Atelip* mutant, and *35S-ScELIP* transgenic seedlings to different light regimes. Panel (**a**), root length in centimeter (cm); panel (**b**), root number; panel (**c**), leaf number; panel (**d**), Total chlorophyll content in milligram (mg) per gram fresh weight. Error bars represent the SD of three biological repeats. ** *p* < 0.01, P-values were obtained from Dunnett’s T3 test comparing the *35-SScELIP* lines with *Atelip* mutant. *P*-values also were obtained from Dunnett’s T3 test comparing the control (normal light condition), red, blue, and UVB light with each other. Asterisk represents statistically significant difference within one group or among specific groups delineated by the brackets.

**Figure 5 ijms-21-01411-f005:**
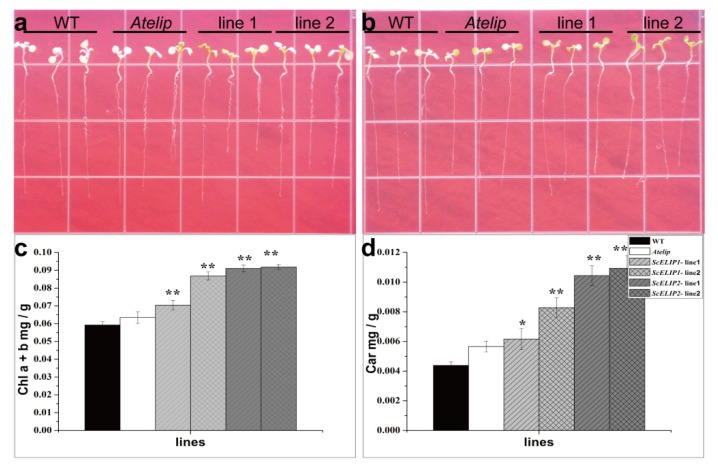
Phenotypic response of Col-0 WT, the *Atelip* mutant, and *35S-ScELIP1* and *35S-ScELIP2* transgenics to high-light irradiation (1500 μmol/m2/s) for 2 h and recovery for 24 h under control conditions. (**a**) Col-0 WT, *Atelip,* and *35S-ScELIP1*; (**b**) Col-0 WT, *Atelip,* and *35S-ScELIP2*; (**c**) Chlorophyll content of Col-0 WT, *Atelip,* and *35S-ScELIP* transgenics after exposure to 2 h of high-light irradiation followed by 24 h of recovery in control conditions; (**d**) Carotenoid content of Col-0 WT, *Atelip,* and *35S-ScELIP* transgenics after exposure to 2 h of high-light irradiation followed by 24 h of recovery in control conditions.

**Figure 6 ijms-21-01411-f006:**
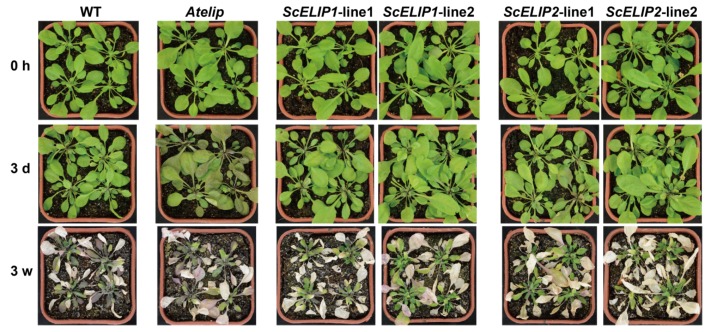
Phenotypic response to high-light exposure for Col-0 WT, *Atelip*, and *35S-ScELIP1,* and *35S-ScELIP2* plantlets. Four-week old *Arabidopisis* lines prior to high-light irradiation treatment (Row 1), at 3 days of high-light irradiation (1000 μmol/m^2^/s) treatment (Row 2), and 3 weeks of continuous high-light irradiation (1000 μmol/m^2^/s) (Row 3).

**Figure 7 ijms-21-01411-f007:**
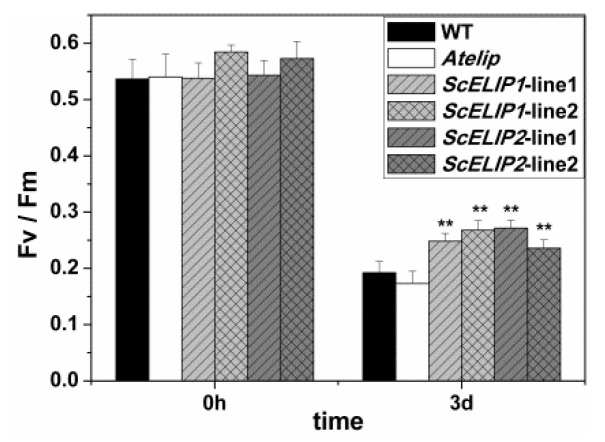
Photosynthetic parameters for WT, *Atelip,* and *35S-ScELIP1* or *35S-ScELIP2* at 0 h (prior to treatment) and 3 days of high-light exposure (1000 μmol/m^2^/s). Error bars represent the SD of three biological repeats. ** *p* < 0.01, P-values were obtained from Dunnett’s T3 test comparing WT and the transgenic lines with the *Atelip* mutant, respectively. Fv/Fm: Maximal quantum yield of PSII, Y(II): Actual quantum yield of PSII, ETR: Photosynthetic electron transport rate.

**Figure 8 ijms-21-01411-f008:**
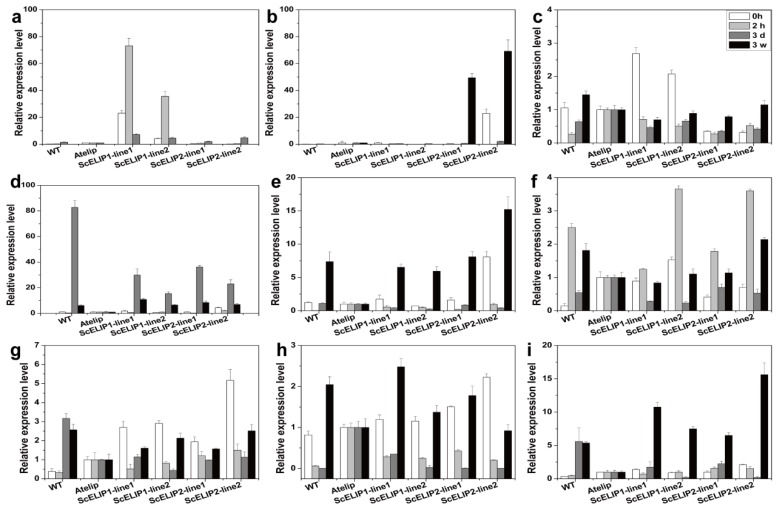
Twelve photosynthesis genes transcript relative abundance assessments in *Arabidopsis thaliana* different genotypes (WT, *Atelip*, transgenic lines) exposed to high-light treatments (1000 μmol/m^2^/s) for various times. Error bars represent the SD of three biological repeats. Relative expression values were obtained from 2^−^^ΔΔ*C*t^ comparing WT and the transgenic lines with the *Atelip* mutant, respectively. *ScELIP1*: The inserted *S. caninervis ELIP1* coding sequence (**a**), *ScELIP2*: The inserted *S. caninervis ELIP2* coding sequence (**b**), *PSBS*: The native *Arabidopsis* PSII CP22 protein gene (*PSBS*) transcript (**c**), *OHP*: The native *Arabidopsis* one-helix protein gene (*OHP*) transcript (**d**), *OHP2*: The native *Arabidopsis* one-helix protein 2 gene (*OHP2*) transcript (**e**), *SEP2*: The native *Arabidopsis* two-helix protein gene (*SEP*) transcript (**f**), *LHCA2*: The native *Arabidopsis* PSI antenna protein gene (*LHCA2*) transcript (**g**), *LHCB4.2*: The native Arabidopsis PSII CP29 antenna protein gene (*LHCB4.2*) transcript (**h**), *PSBD*: The native *Arabidopsis* PSII D2 protein gene (*PSBD*) transcript (**i**).

**Table 1 ijms-21-01411-t001:** Properties of cloned ScELIPs ^1^.

Gene Name	*ScELIP1* ^2^	*ScELIP2* ^3^
Length of coding sequence (bp)	995	904
ORF (bp)	711	624
Predicted number of amino acids	236	207
Predicted molecular mass (kDa)	25.68	21.69
Predicted pI	9.33	9.57

^1^ The early light-induced protein of *Syntrichia caninervis*. ^2^ The early light-induced protein gene 1 of *Syntrichia caninervis*. ^3^ The early light-induced protein gene 2 of *Syntrichia caninervis.*

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
