# Peer review of "Functional Aspects of Early Light-Induced Protein (ELIP) Genes from the Desiccation-Tolerant Moss *Syntrichia caninervis"

_ijms, 2020, doi:10.3390/ijms21041411_

Round 1
Reviewer 1 Report
The manuscript reports the characterization of two ELIP genes isolated from the desiccation tolerant moss Syntrichia caninervis . The genes were characterized in two ways. First the expression of the transcripts was measured in response to different light stresses and different light qualities. These experiments showed a differential transcript accumulation for both genes analysed. The response of the genes is not identical but similar. In the second approach an Arabidopsis mutant, in which the AtElip2 gene has been inactivated, was complemented with either moss gene. The transgenic Arabidopsis lines overexpressing the moss gene were examined for the question whether the moss genes can complement the mutant phenotype. It was shown that the moss genes were able to complement the mutant phenotype, although not in all cases to 100%. The experiments show that the ELIP genes from the desiccation tolerant moss Syntrichia caninervis contribute to protection under light stress. Thus this work provides more experimental evidence confirming the protective function of ELIP genes.
Nevertheless there are some points which need to be addressed:
Lines 107 to 113: it is not clear why the unigenes 40121 and 68225 were chosen out of the unigenes available in the genome of Syntrichia caninervis? Were the remaining genes also characterized? An expansion on these genes would complement this manuscript nicely.
Was the protein expression tested in Arabidopsis overexpressing the moss genes? This would be interesting to know, because the protein analysis could provide an explanation why the complementation did not always work fully.
The manuscript is generally clearly written. There are a few errors e.g.
Line 34 “provides”
Lines 332 to 342 contain several grammatical errors mainly related to singular and plural e.g. “bryophytes is a “ (line 335). Line 341 “transcript abundance were”
Line 346 “ELIPs transcripst”
“excepting one kind of ferns” is not clear.
Line 333 omit”And”
The manuscript should be checked carefully
Author Response
Dear reviewer:
Thank you very much for your comments and suggestion.
As suggested, I have checked manuscript carefully, I have corrected tense, spelling mistake, singular and plural forms that mentioned in reviewer report and other errors.
Replies to Reviewer 1:
it is not clear why the unigenes 40121 and 68225 were chosen out of the unigenes available in the genome of Syntrichia caninervis? Were the remaining genes also characterized? An expansion on these genes would complement this manuscript nicely.
Answer: We identified 26 Unigenes as ScELIPs in the reference transcriptome, 6 of which have complete open reading frames (ORFs): Unigenes 13021, 40120, 40121, 68225, 16576, and 8044. Unigene 16576 and 8044 each contain 94 predicted amino acid that were of short length to known ELIPs (see Supplementary Table S1). The predicted pI of Unigene 13021 was 5.58 which significantly lower than other ScELIPs Unigenes. Unigenes 40121 and 68225 each contain ORFs that were of comparable length to known ELIPs and encode a putative primary amino acid sequence that matched known ELIP proteins. This is one reason why we choose Unigenes 40121 and 68225 as research genes. I have added a table about properties of 6 Unigenes of ScELIPs as Supplementary Table S1.
Was the protein expression tested in Arabidopsis overexpressing the moss genes? This would be interesting to know, because the protein analysis could provide an explanation why the complementation did not always work fully.
Answer: I have not tested ScELIPs protein expression in Arabidopsis overexpressing the moss genes for now, I would like to test it, because my experiment about the function of ScELIPs under drought, salt, cold and heat haven't be finished.
Thank you very much for your positive and constructive comments and suggestions, it is very important for me.
With kindest regards,
Yours Sincerely,
Xiujin Liu
Reviewer 2 Report
Dear Authors,
I have indicated my corrections and concerns on the attached file itself.

Author Response
Dear Reviewer:
Thank you very much for your comments and suggestion.
As suggested, I have revised my manuscript as possible as I can. All of my figures were low resolution and the letters were smudgy, actually, my figures were sufficiently high resolution (more than 300 dpi), but these figures became blurry pictures when submited to system online. So I resubmit PDF file this time, figures quality is OK now.Most of my figure legend were incomplete, I have supplemented. I have corrected all "Col-O" to "Col-0".
Replies to reviewer 2:
Which light? White lighe, bule or red please specify.
Answer: The transcript abundance for both ScELIPs increase with increasing white light intensity. We used cool-white fluorescent light lamp as light source for high light treatment (as mentioned in material and method), and I have added white light in line 164.
Please perform statistical analysis, such as Tukeys-HSD test and indicate the letters of significance on top of bar. Please indicate data represent mean of how many replicates in the legend.
Answer: In figure 2, I want to display transcript abundance of ScELIPs under different light quality, intensity and different time point. I haven’t compared significance between ScELIP1 and ScELIP2, or different time point. If it’s necessary, I will add it.
Given the significant differences in the levels of the transcript in the transgenic lines, I am wondering whether there is visible phenotype could be observed by the authors. Authors should also mention whether there is a phenotypic differences observed in Arabidopsis mutant lines ELIPS and WT. Does the over-expression of the ScELIPS complements the phenotype?
Answer: Although the transcript abundance of over-expression ScELIPs transgenic lines were higher than Atelip mutant, there isn’t visible phenotype difference between the Col-0 WT, the Atelip mutant and the ScELIPs transgenic lines in normal condition. Over-expression of ScELIPs can complement the function of ELIP in Atelip mutant under high light stress. I have complete this part at page 6 line 178-181.
Authors need the represent the data for the seed germination of the tested genotype under normal white light conditions.
Answer: In Figure 3a, b represent the germination rate of ScELIPs transgenic lines under normal white light condition, there isn’t obviously difference (line 188). The germination phenotype under all kinds of light including white light displayed in supplementary figure S8, there aren’t visible diffenence between three genotypes under normal white condition.
The highlighted statement is not reflected in the supplemntary figure 9. I find no significant reduction in the number of leaves albeit all of them exhibit increase length of the petiole.
Answer: Red light as single light source lead to long petiole, yellow leaf, and the number of leaves (Figure 4c) and overall root lengths (Figure 4a) were reduced in comparison to the controls. I have revised statistic analysis in Figure 4, it’s correct, while the picture I choosed in the supplemntary figure 9 was not representative, so I choosed typical pictures in the supplemntary figure 9.
Please mention the control condition.
Answer: The control condition is 22 ± 2ºC with a 14 h light/ 10 h dark cycle and 60–75 % relative humidity and light at PPFD of 100 μmol/m2/s as write in material and method, and I have added it at line 246.
Improve the quality of the pictures. Indicate the carotenoid content in the high light treated seedlings. In the figure 5a. Please indicate the genotype on the top of the instead of the bottom of the seedlings. The length of the roots is masked due the placement of the legend.
Answer: I have improved figure 5 according to suggestion, and added Figure 5d about the carotenoid contents in the high light treated seedllings, the carotenoid contents were corresponding with chlorophyll contents.
This statement is not clear. In my view there merely a subtle differences rather than a significant. Authors have added the data to the figure. I think authors meant 3weeks here….
Answer: Figure 6 is low-definition image in this PDF file, actually, I have repeated this experiment many times, Atelip mutant appears to be more severe than other genotype under high light stress, Atelip mutant became more dark green or purple than other genotype every time. Although plant were still green after 3 day high light treatment, the Fv/ Fm values were so low as to be unreliable and/or undetectable so this data was not added to the analysis, so I haven’t measured Fv/ Fm in 3weeks.
Thank you for your detailed review, I hope you can give me one more chance, It's very important for me.
With best wishes,
Yours sincerely,
Xiujin Liu
Round 2
Reviewer 2 Report
Dear Authors,
The revised version that you have submitted is significantly improved and clear now.
a)However, it is pertinent that you should include the statistical analysis for all the panels in figure2 for benefit of the readers and scientific relevance.
b) In the figure5 panel a) b) and c) are duplicated please delete the one where the legend are marked below the seedlings.
Please pay attention to the revisions before you resubmit.
c) Please perform statistical analysis and significance test of the transcripts abundance of the photosynthetic genes in the figure 8. It is essential.
FYI: For future manuscripts, it is advisable to take the pictures of the seedlings in the Petri dishes with black back ground for sake of clarity.
Author Response
Dear Reviewer:
Thank you very much for your pertinent suggestions on our manuscript, and giving me one more chance to revise and perfect our manuscript.
As suggested, we have revised our manuscript and completed figures. I have added significant analysis in figure 2 and 8, and completed figure legend. I have addressed the comments, and the amendments are highlighted in red in the revised manuscript. I hope that the revision is accptable and look forward to hearing from you soon.
With best wishes,
Yours sincerely,
Xiujin Liu
Response to reviewer 2:
a)However, it is pertinent that you should include the statistical analysis for all the panels in figure2 for benefit of the readers and scientific relevance.
Answer: I have supplemented the statistical analysis in figure 2 and completed figure legend. P-values were obtained from Dunnett’s T3 test comparing different time point and treatment with control.
b) In the figure5 panel a) b) and c) are duplicated please delete the one where the legend are marked below the seedlings. Please pay attention to the revisions before you resubmit.
Answer: I made a mistake, I think that I should keep every change in manuscript, so I haven’t deleted the old version of figure 5. For now, I have deleted the old version and kept the new one.
c) Please perform statistical analysis and significance test of the transcripts abundance of the photosynthetic genes in the figure 8. It is essential.
Answer: I can’t add statistical analysis result in the old version figure 8, I have redraw figure 8, and supplemented statistical analysis in the new edition figure 8, I hope that the new edition figure 8 is suitable, Please see attachment.
FYI: For future manuscripts, it is advisable to take the pictures of the seedlings in the Petri dishes with black back ground for sake of clarity.
Answer: I sorry that I have took all photos of the seedling in the Petri dished with red cloth as back ground. Should I transform back ground to black with Photoshop? I am not sure if it is OK?
